# Characterization of Bioactive Compounds in Spent Mushroom Substrate: A Metabolomic Perspective on Its Untapped Potential

**DOI:** 10.3390/foods15010109

**Published:** 2025-12-30

**Authors:** Lanxin Zhang, Irwin Kee-Mun Cheah, Yuyun Lu, Dejian Huang, Alvin Eng Kiat Loo

**Affiliations:** 1Department of Food Science and Technology, Bezos Centre for Sustainable Protein, National University of Singapore, Singapore 117542, Singapore; lanxinz@u.nus.edu (L.Z.); fstluy@nus.edu.sg (Y.L.); dejian@nus.edu.sg (D.H.); 2National University of Singapore (Suzhou) Research Institute, Suzhou 215123, China; 3Department of Biochemistry, Yong Loo Lin School of Medicine, Centre for Life Sciences, National University of Singapore, Singapore 117456, Singapore; bchickm@nus.edu.sg

**Keywords:** spent mushroom substrate, ergothioneine, food waste, metabolomics, LC-QTOF-MS/MS

## Abstract

Ergothioneine is an emerging natural product with anti-aging activity and highly sought after. In this study, we examined spent pink and pearl oyster mushrooms substrate for ergothioneine levels. However, we found that the ergothioneine contents were quite low, ranging from 1.66 to 9.14 μg·g^−1^ and 2.00 to 10.38 μg·g^−1^, respectively, with no significant increase compared to the unused substrate. Untargeted metabolomic analysis revealed that many low-molecular-weight compounds, including polyphenols, were reduced in spent substrates after mushroom cultivation, likely due to absorption, degradation, or utilization by the fungal mycelium. Interestingly, the spent substrates from both mushroom species were found to be enriched in certain compounds, particularly cerebroside B and ganoderenic acid D. These compounds are potentially valuable products that may be extracted from SMS. However, further targeted analytical validation is required to confirm the identity of and quantify these compounds.

## 1. Introduction

Mushrooms have long been valued not only for their culinary applications but also for their nutritional and medicinal properties. They are rich in bioactive compounds such as polysaccharides, polyphenols, sterols, terpenoids, and essential minerals, which have been associated with antioxidant, anticancer, immunomodulatory, antidiabetic, neuroprotective, and other health-promoting effects [1,2,3,4].

The edible mushroom industry has become the fifth most significant agricultural sector [5]. However, large-scale mushroom cultivation results in the generation of substantial amounts of spent mushroom substrate (SMS). On average, for each kilogram of mushrooms produced, approximately five kilograms of waste are generated [6]. It is projected that by 2026, mushroom cultivation will produce approximately 104 million tons of waste globally [7]. The accumulation of such waste poses both environmental and economic challenges.

In response to these environmental and economic challenges, there has been growing interest in the value-added utilization of SMS. It has been reported that SMS contains a variety of bioactive compounds and essential nutrients, including antioxidants, lignocellulolytic enzymes, and minerals like potassium and magnesium [8,9,10]. Consequently, its reuse has been studied in agriculture, like soil amendment [11,12], upcycled crop substrates [13], animal feed [14], environmental remediation [15,16], and even food preservation [17].

Among the diverse metabolites present in mushrooms, notable differences have been observed in their amino acid profile between species [18]. Ergothioneine (EGT) is a sulfur-containing amino acid synthesized by certain fungi and bacteria. Mushrooms are recognized as the richest dietary source of EGT, although its content varies greatly with species [19]. EGT exhibits strong antioxidant and cytoprotective properties and has been associated with neuroprotection [20], cardiovascular health [21], maternal health during pregnancy [22], and aging-related outcomes [23,24,25]. It is taken up by the human body via the OCTN1 transporter and is suggested to accumulate in all tissues but preferentially to those predisposed to oxidative stress, including the brain, suggesting a role in neurodegenerative disease prevention [20,26,27].

Animals are incapable of synthesizing EGT endogenously; hence, they must obtain it from their diet, with mushrooms recognized as the richest dietary source. Given that some metabolites, like polysaccharides, sterols, and other compounds, found in fruiting bodies can also be detected in SMS [28], we hypothesized that SMS may also contain EGT and other valuable bioactive compounds. Furthermore, given that mushrooms are usually cultivated in bags of substrates commercially and each batch of cultivation bag may fruit more than once, we hypothesize that the amount of these metabolites in the substrate may also change with the harvest cycles. A previous study has recovered EGT as a desirable product from mushroom waste [29]. However, to the best of our knowledge, no studies have systematically quantified the EGT content in SMS.

This study aims to analyze the potential bioactive compounds in SMS across different harvest cycles. By exploring the metabolites in SMS, this work contributes to the sustainable valorization of SMS and its potential applications.

## 2. Materials and Methods

### 2.1. Chemicals, Standards, and Reagents

L-ergothioneine (EGT), L-hercynine, ergothioneine sulfonate (ET-SO_3_H), S-methyl-ergothioneine (S-Met-L-Erg), and the isotopically labeled internal standards L-ergothioneine-d9 and L-hercynine-d9 were obtained from ERGOLD (Paris, France). Gallic acid, sodium carbonate, and Folin–Ciocalteu phenol reagent were supplied by Sigma-Aldrich (St. Louis, MO, USA). HPLC-grade acetonitrile (ACN), methanol, and mass spectrometry-grade formic acid (FA) were purchased from Fisher Scientific (Hampton, NH, USA). LC-MS-grade water and ACN were sourced from Aik Moh Paints & Chemicals Pte Ltd. (Singapore).

### 2.2. Sample Collection

The growth kits for *Pleurotus ostreatus* (pearl oyster) and *Pleurotus djamor* (pink oyster) were purchased from Bewilder (Singapore), and their main component was oak sawdust. Mushroom cultivation was initiated by creating X-shaped slits on the cultivation bags, and the bags were cultivated in a horticultural tent. The relative humidity was maintained at 80–90% using a humidifier (Bear JSQ-C50Q1, Bear Electric Appliance Co., Ltd., Foshan City, China) and regular misting with water sprays. Electric fans (JYF-110-13, China) were used to ensure continuous air circulation. A total of 15 bags for each species were cultivated, and 5 bags were harvested for each time point.

Upon maturation, the fruiting bodies were harvested by carefully detaching them from the substrate. For the collection of spent mushroom substrate, two types of samples were obtained, as shown in Figure 1. The first sample consisted of SMS located near the surface, directly beneath the fruiting bodies, which contained a higher density of mycelium. The second sample was taken from the center of each bag of the cultivation kit. After harvest, both the fruiting bodies and the SMS were frozen at −20 °C overnight and subsequently freeze-dried (Buchi, Singapore) and ground into powder. The samples were then stored at −20 °C prior to analysis. In addition to the cultivated samples, substrates that were derived from the same batch as the growth kits but had not been used for mushroom cultivation were also obtained from the manufacturer for analysis. These are termed unused substrates in this paper.

### 2.3. Detection of EGT

#### 2.3.1. Sample Preparation

Samples (20 mg) were placed into a 5 mL centrifuge tube with 5 mL of ultrapure water. The mixture was vortexed for 30 s and left to stand for 30 min. The sample was centrifuged at 18,000× *g* for 5 min using a Beckman Microfuge 16 microcentrifuge (Beckman Coulter, Brea, CA, USA), and the supernatant was collected and diluted ten-fold before being filtered for LC-MS/MS analysis.

#### 2.3.2. Liquid Chromatography–Tandem Mass Spectrometry (LC-MS/MS) Analysis

The LC-MS/MS analytical method was adapted from a previously established protocol [27]. The system comprised an Agilent 1290 ultra-high-performance liquid chromatography system (UPLC) coupled to an Agilent 6460 ESI tandem mass spectrometer. Samples were maintained at 10 °C in the autosampler, and 2 µL of each processed sample was injected into a Cogent Diamond-Hydride column (4 µm, 150 × 2.1 mm, 100 Å; MicroSolv Technology Corporation, Leland, NC, USA) held at 40 °C. The mobile phase consisted of Solvent A (0.1% FA in ultrapure water) and Solvent B (0.1% FA in ACN). Chromatographic separation was achieved using a gradient elution at a flow rate of 0.5 mL/min. The gradient began with 10% Solvent B for 2 min, followed by an 8 min increase to 60% Solvent B for elution of ET-SO_3_H and EGT. Subsequently, Solvent B was increased to 100% for the elution of hercynine and S-Met-L-Erg, maintained for 2 min, and then returned to 10% for 3 min to re-equilibrate the column. The total run time was 15 min, and retention times were as follows: ET-SO_3_H, 3.63 min; EGT, 4.68 min; hercynine, 8.89 min; and S-Met-L-Erg, 9.09 min.

Mass spectrometric detection was performed in positive ion mode using electrospray ionization and multiple-reaction monitoring for the quantification of target ions. The capillary voltage was set to 3200 V, with a gas temperature of 350 °C. Nitrogen was used as the nebulizing gas at a pressure of 50 psi, with a gas flow rate of 12 L/min, and as the collision gas in ultra-high-purity form. The precursor-to-product ion transitions, along with the corresponding fragmentor voltages (V) and collision energies (eV), were as follows: EGT (230.1 → 186.2, 106 V/9 eV), EGT-d9 (239.1 → 195.2, 110 V/10 eV), hercynine (198.2 → 95.2, 95 V/21 eV), hercynine-d9 (207.2 → 95.2, 95 V/22 eV), S-Met-L-Erg (244.1 → 141, 92 V/17 eV), and ET-SO_3_H (278.1 → 154.1, 120 V/15 eV).

### 2.4. Total Phenolic Content

#### 2.4.1. Sample Preparation

A one-hundred-milligram SMS sample was weighed into a microfuge tube, and 1 mL of ultrapure water was added. The mixture was ground using TissueLyser (QIAGEN, Singapore) at 25 Hz for 30 s and left to stand for 30 min. The sample was centrifuged at 18,000× *g* for 5 min, and the supernatant was collected. This process was repeated two more times, and the supernatants were combined for subsequent analysis.

#### 2.4.2. Determination of Total Phenolic Content

The total phenolic contents (TPC) in the extracts were quantified using the Folin–Ciocalteu method [30,31]. Briefly, 100 μL of diluted Folin–Ciocalteu’s phenol reagent was added to 20 μL of extracts on a 96-well plate, followed by the addition of 80 μL of a 7.5% Na_2_CO_3_ solution. The mixture was incubated at room temperature for 30 min, and then the absorbance was subsequently measured at 750 nm using a PowerWave XS2 Microplate Spectrophotometer (BioTek; Mason Technology, Dublin, Ireland). The TPC of the extract was expressed as mg gallic acid equivalent (GAE)·g^−1^ dry mass.

### 2.5. Elemental Analysis

The elemental composition (carbon, hydrogen, nitrogen, and sulfur) of the spent mushroom substrate was determined to assess its basic biochemical characteristics [32]. The analysis was performed using a ThermoFisher Scientific FlashSmart Elemental Analyzer (Waltham, MA, USA), calibrated with a sulfanilamide standard.

### 2.6. Untargeted Metabolomics

#### 2.6.1. Sample Preparation

A five-hundred-milligram SMS was added to a 5 mL centrifuge tube with 5 mL of 80% methanol. The mixture was vortexed for 30 s and allowed to stand for 30 min. The sample was then centrifuged at 18,000× *g* for 5 min, and the supernatant was collected and incubated overnight to facilitate protein removal. After incubation, the sample was centrifuged again at 18,000× *g* for 5 min. A 2 mL aliquot of the supernatant was collected, dried, and subsequently resuspended in 2 mL of ultrapure water. Finally, the resuspended solution was filtered through a 0.22 µm hydrophobic PTFE membrane, and the filtrates were used for LC-QTOF-MS analysis.

#### 2.6.2. Liquid Chromatography–Quadrupole Time-of-Flight Mass Spectrometry (LC-QTOF-MS) Analysis

The prepared samples were analyzed in a randomized order through an ultraperformance liquid chromatography (UPLC) system coupled to a Waters Xevo G2-XS quadrupole time-of-flight mass spectrometer (QTOF-MS), Nonlinear Dynamics, Waters, Newcastle, UK, equipped with an electrospray ionization (ESI) source. The QTOF-MS analysis was performed following a method adapted from prior work on spent coffee grounds [33]. Chromatographic separation was performed on a Waters Cortecs T3 column (3 mm × 100 mm, 2.7 μm) maintained at a constant temperature of 50 °C. The mobile phase consisted of 0.1% FA in water (solvent A) and 0.1% FA in ACN (solvent B), delivered at a flow rate of 0.5 mL/min under gradient elution. The gradient program began with a mobile phase containing 95% Solvent A during the first minute. Between 1 and 3 min, the proportion of Solvent A decreased linearly from 95% to 65%, followed by a further reduction to 20% from 3 to 4 min. From 4 to 9 min, Solvent A was gradually reduced to 5% and maintained at this level for three minutes. At 12.1 min, the proportion of Solvent A was restored to 95% and held constant until the end of the run at 15.1 min. The mass range was set from 100 to 1500 *m*/*z*. The ESI source operated with a capillary voltage of 3.0 kV and a sampling cone voltage of 40 V, while the source offset was maintained at 80 V. The ion source temperature was 120 °C, and the desolvation temperature was 500 °C. Nitrogen was used as the cone gas at a flow rate of 50 L/h and as the desolvation gas at a flow rate of 1000 L·h^−1^. Collision energy was maintained at 6.0 V throughout the analysis.

#### 2.6.3. Data Processing

For the LC-QTOF-MS/MS analysis, raw data were processed and visualized using Progenesis QI v3.0 software (Nonlinear Dynamics, Waters, Newcastle, UK). The analysis included principal component analysis (PCA), one-way analysis of variance (ANOVA) with multiple testing correction (FDR threshold < 0.05), and paired *t*-tests within groups. Fold change (FC) analysis was performed to identify metabolites with significant alterations across groups. The identified metabolites were subsequently mapped to metabolic functions using resources such as FOODB, ChEBI, and PubChem databases. Metabolite identities were assigned based on accurate mass, isotopic pattern, and database matching within Progenesis QI.

### 2.7. Statistical Analysis

Data processing was carried out using the ANOVA method in OriginPro 2021b (OriginLab Corporation, Northampton, MA, USA). Statistical significance was defined as a *p*-value less than 0.05.

## 3. Results

### 3.1. Yield of Mushroom Fruiting

Each mushroom cultivation kit weighs approximately 1 kg, and the yields are shown in Figure 2. Pearl oyster mushroom could be harvested twice, with a mean yield of 250.3 g and 64.5 g at the first and second harvests, respectively. In our study, pink oyster mushroom could be harvested three times, with yields of 178.3 g, 89.6 g, and 31.4 g, respectively. There was a significant decline in the yields with each subsequent harvest, which was expected.

### 3.2. Content of EGT and Related Metabolites

We categorized the SMS based on the type of mushroom (pink and pearl oysters), site of harvest (surface or core), and the number of harvests that have been obtained from the SMS. Surprisingly, the unused substrate already possessed basal levels of EGT, and none of the SMS samples had higher EGT levels than the unused substrate. In fact, several of the substrates had EGT content even lower than the unused substrate (Table 1).

Next, we observed that there was a general trend that the surface of the substrate does contain more EGT than the core, and the difference was statistically significant for several time points. However, we did not observe a clear trend in the number of harvest cycles and the EGT content. There was an increase in EGT with the harvest number for pearl oyster mushrooms, but the same trend was not observed for pink oyster mushrooms.

Similarly, there was no clear trend in the effect of harvest cycle on the EGT content of the fruiting bodies (Table 2). There was an increase in ergothioneine with harvest cycle for pink oyster mushrooms but not pearl oyster mushrooms. The pearl oyster mushrooms did not fruit for a third time.

Overall, the results indicated that SMS is not a good source of EGT, with levels comparable to or lower than the unused substrate. The EGT content of SMS is also significantly lower than that of the fruiting body.

The content of hercynine, the biosynthetic precursor to EGT, largely mirrored the trend of EGT during mushroom growth and across SMS types (Table 3). Specifically, the SMS of pink oyster mushroom demonstrated a consistent decline in hercynine levels with each harvest, while the SMS of pearl oyster mushroom displayed an increasing trend. Moreover, the hercynine content on the surface of SMS consistently exceeded that of the core, underscoring a spatial concentration gradient within a bag of SMSs. The hercynine content of the mushroom fruiting bodies is shown in Table 4. In both pink and pearl oyster mushrooms, a decrease in hercynine concentration was observed from the first to the second harvest, dropping from 835.29 to 738.80 μg·g^−1^ and further to 150.11 μg·g^−1^ in pink oyster mushrooms and from 760.03 to 543.96 μg·g^−1^ in pearl oyster mushrooms. This trend aligned with the EGT content patterns, indicating a parallel fluctuation across successive harvests.

Ergothioneine sulfonate (ETSO_3_H), the oxidized form of EGT, was detected in both SMS and mushroom fruiting bodies. While no consistent trends were observed between different harvests or different types of SMS, significant variations were noted between mushroom species, particularly during the second and third harvests (Appendix A).

### 3.3. TPC Values

As SMS has been reported to be a potential source of antioxidants [9], the TPC of SMS was measured as an indicator of total antioxidant content. However, similarly to the findings for EGT, the TPC of SMS was comparable to that of the unused substrate (Table 5). Across different harvests, a decreasing trend in TPC was observed in pink oyster SMS but not the pearl oyster SMS (Table 6). On the other hand, it was observed that the TPC for fruiting bodies reduced with the number of harvests.

Generally, SMSs from different sites (surface versus core) exhibited variability in EGT, hercynine, and TPC. However, no significant differences were observed in TPC levels across these locations. This suggests that the metabolic composition of the substrate is influenced by spatial and environmental factors during mushroom cultivation.

As the TPC was measured in an aqueous extract, we determined the CHNS composition to rule out the possibility that the changes in TPC with harvest cycles were due to changes in protein and amino acid content. However, there were no significant changes in the CHNS composition of the substrate with harvest cycles (Appendix A). There was a clear, albeit not statistically significant, decreasing trend in the protein content of the mushroom fruiting body with increasing harvest cycles. This may at least partially explain the decrease in TPC observed in mushroom fruiting bodies with increasing harvest cycles.

### 3.4. Untargeted LC-QTOF-MS/MS Metaboloite Analysis

#### 3.4.1. Global Profile of Untargeted Metabolomics in SMS and Unused Substrate

A comprehensive untargeted metabolomic analysis was performed on the SMS using LC-QTOF-MS/MS to compare the difference between SMS and unused substrate. A total of 5181 unique compounds were detected across the samples. To ensure data reliability and relevance, strict filtering criteria were applied: compounds with an ANOVA *p*-value < 0.05, a coefficient of variation (CV) < 30%, and a maximum fold change > 2 when compared to unused substrate were retained. Unidentified compounds (those not annotated in available databases, METLIN MS/MS Library and ChemSpider, or with missing data) were excluded [34]. This process resulted in a subset of 2048 compounds for further analysis.

A PCA was carried out for all the SMS samples. The first two principal components (PCs), PC1 and PC2, explained 33.46% and 21.47% of the variance, respectively. The PCA showed that the unused substrates cluster significantly away from all the SMSs. Core and surface SMSs tend to separate themselves on the PCA plot as well, although overlaps can still be observed, particularly between the pink oyster core and surface SMSs (Figure 3). The quality control (QC) sample, which consists of equal volumes from each sample, was centered in the PCA, indicating that the metabolomic analysis is indeed representative. We also analyzed the results based on the number of harvests, as illustrated in Appendix A. Similarly to the results reported for EGT and related metabolites, there was significant overlap between the SMSs from the first, second, and third harvests. Altogether, the results indicated that most of the changes in the SMS composition happened before the first harvest.

Subsequent comparative analyses between the SMS and unused substrate were conducted to investigate the metabolic changes during mushroom cultivation. The variable importance in projection (VIP) was used to assess the contribution of each variable to the partial least squares discriminant analysis (PLS-DA) model. Variables with VIP ≥ 2 and *p*-value < 0.05 (from independent-samples *t*-test) were considered significant metabolites. Several di- and tripeptides were detected but were excluded as it was expected that there would be protein hydrolysis of the substrate during mushroom cultivation. Only low-molecular-weight compounds with statistically significant differences were retained for further analysis. A total of 58 compounds were retained for further analysis.

Figure 4 illustrates the distribution of these 58 compounds across the four types of SMSs. Among them, 38 compounds were commonly detected in all groups and were subjected to further analysis; the details are available in Appendix A. The remaining 20 compounds, which are unique to certain SMS groups, are listed in Appendix A.

#### 3.4.2. Identification of Compounds

As shown in Figure 4, most of the compounds identified were present in all the SMSs. These compounds were further analyzed in the form of a heatmap (Figure 5). Notably, compounds such as steroids and vitamin D derivatives, certain terpenoids (minabeolide), lipids and lipid derivatives (3-[(3-hydroxytridecanoyl)oxy]-4-(trimethylammonio)butanoate), and several other metabolites (e.g., cerebrosides) were elevated in the SMS compared to the unused substrate. In contrast, polyphenols, the majority of lipids and their derivatives, and tetrapyrroles and related cofactors were generally reduced in the SMS. However, no consistent trend of increasing or decreasing compound accumulation was observed with successive mushroom harvests.

Several compounds were identified exclusively in one type of SMS but not the others, as detailed in Appendix A. In spent pearl oyster mushroom substrate, these unique compounds identified include oleamide, 12-oxo-13-hydroxy-9Z-octadecenoic acid, and 3-PT-phosphatidylinositol (3,4,5)-trisphosphate (1,2-dioctanoyl). In contrast, the spent pink oyster mushroom substrate was characterized by the presence of mangiferin, 4,5-dimethoxy-1,2-benzenedicarboxylic acid, 1-linoleoylglycerophosphocholine, adenosine, and stachyose.

## 4. Discussion

EGT and its related metabolites were quantified in mushroom fruiting bodies and SMS in this study. To the best of our knowledge, this is the first study to quantify the levels of EGT and related metabolites in SMS. There were variations in EGT content across different harvests cycles; however, no consistent pattern was identified. This may be due to differences in external factors such as environmental conditions or physiological stress responses, which may still be present in the experiment despite our best attempt at controlling their growth conditions.

In this study, we also observed spatial complexity in SMS, with EGT and hercynine exhibiting higher levels in the surface substrate compared to the core substrate. Additionally, hercynine displayed divergent temporal trends between mushroom types. These patterns underscore the spatial and temporal complexity of metabolite dynamics within SMS.

Interestingly, while EGT was present in the unused substrate, ETSO_3_H and hercynine were not detectable in the unused substrates and were only present in the SMS. Their presence is likely a consequence of fungal metabolism. However, the potential health benefits of these compounds remain unexplored.

It has been suggested that SMS could serve as a potential source of antioxidants [9]. However, we found no statistically significant enrichment of antioxidants in the SMSs compared to the unused substrates, which also aligns with our findings for EGT. One possible explanation is that mushrooms actively absorb antioxidant compounds or their precursors from the substrate during growth and retain them within their tissues. A previous study has shown that EGT biosynthesis in mushrooms such as *Flammulina velutipes* is tightly regulated and depends on the uptake of specific amino acid precursors from the substrate, followed by intracellular synthesis and storage [35]. Furthermore, analysis of post-harvest mushrooms has demonstrated that EGT is highly stable within fungal tissue and only degrades significantly under extreme heat or prolonged boiling, indicating that it is not readily secreted into the environment during cultivation [36]. These findings support the view that EGT and other nutrients are efficiently sequestered in the fungal biomass and are minimally released into the SMS. In addition, variations reported in the literature regarding SMS antioxidant capacity likely stem from differences in fungal species, substrate formulations, and extraction solvent polarity, all of which can substantially influence TPC measurements across studies [37,38].

The PCA plot of the untargeted metabolome analysis (Figure 3) revealed that the composition of SMS was easily distinguishable from unused substrates; however, there was significant overlap between SMSs from different sampling locations. We also conducted PCA based on the number of harvest cycles (Appendix A) and also found significant overlaps between samples from different harvest cycles. This indicates that mushroom cultivation by itself was the largest factor affecting changes in SMS composition. The number of harvest cycles and the location of the substrate played a less significant role. While counterintuitive, these results are expected. The growth cycle of cultivated mushrooms starts with the inoculation of the mycelium, which goes on to colonize the entire bag of substrates. After the entire bag of substrates has been colonized, fruiting is triggered by increasing exposure to oxygen. Hence, by the time the mushrooms and substrates are ready for collection, any growth and metabolite production in the spent substrate may have reached a steady state plateau [39,40].

As illustrated in Figure 5, most of the compounds detected, particularly the polyphenols, were present at lower concentrations in the SMS. This finding contradicts earlier reports that SMS could be a source of polyphenols [9] but aligns with findings that mushrooms lack key the enzymes required for flavonoid biosynthesis [41]. Given that mushrooms are unable to produce flavonoids, the decrease in polyphenols observed is most likely due to fungal degradation of the polyphenols originally present in the substrate. The levels of lipids and lipid derivatives were lower in the SMS compared to the unused substrate, which is consistent with the notion that fungi can assimilate exogenous lipids during growth [42]. Tetrapyrroles, which play essential roles in light signaling, respiration, photosynthesis, and programmed cell death, also showed a decrease in abundance during mushroom cultivation [43]. In contrast, terpenoid levels exhibited both increases and decreases. This variability may be attributed to their role as metabolites, which are not essential for primary growth and development but are instead involved in the plant’s defense against biotic and abiotic stresses [44].

Despite SMS not being a rich source of EGT, our untargeted analysis highlighted other valuable bioactive compounds. One such compound is putatively identified as ganoderenic acid D, a triterpenoid known for its anti-inflammatory and hepatoprotective effects [45,46]. While ganoderenic acids are most associated with *Ganoderma* spp., they have also been recently reported in traditional Chinese medicine preparations [47]. This study shows that SMS from *Pleurotus* spp. may also contain ganoderenic acid D.

Several vitamin D-related derivatives were detected. Although ergocalciferol (vitamin D_2_) itself was not observed, the presence of its metabolic products aligns with previous reports of mushrooms as a non-animal source of vitamin D [3,48]. It is plausible that most of the vitamin D may have been converted into these metabolites in the SMS but this would need to be further verified.

Cerebrosides are found across plants, fungi, and animals. They have been isolated from African medicinal plants and are known for their potent anti-inflammatory properties, along with weak antifungal and antimycobacterial activities, and have also been associated with improving the skin’s water-retention capacity [49]. Minabeolide is commonly found in soft corals and has also been identified in *Withania aristate* (Aiton) Pauquy, a medicinal plant endemic to the North African Sahara [50]. It has demonstrated both cytotoxic activity against cancer cells and anti-inflammatory effects [51]. The compounds that are unique to either the pink or pearl oyster SMS are shown in Appendix A, and some of them have potential health benefits. For instance, stachyose is a prebiotic oligosaccharide, supports beneficial gut microbiota, and may promote intestinal health [52]. Phosphopantetheine is a prosthetic group for acyl carrier proteins in fatty acid biosynthesis, with potential antioxidant properties and anti-atherogenic roles [53,54]. Adenosine, involved in energy transfer and cellular signaling, may also enhance testicular activity and counteract fatigue [55]. These compounds have demonstrated potential health benefits and may therefore warrant further analysis in SMS. Although SMS may not be suitable for nutraceutical recovery of EGT, it may serve as a low-cost source of potential bioactive compounds such as ganoderenic acid D and cerebroside B.

A key limitation of this study is that the metabolites annotations were assigned based solely on accurate mass and database matching without MS/MS spectral confirmation or validation with analytical standards. Further targeted analytical validation is required to verify that SMS is a source of these bioactive compounds.

## 5. Conclusions

While EGT was the primary compound of interest, it was present at low levels even in the unused substrate. The EGT levels in the SMS either remained unchanged or showed a slight decrease. Similar trends were also shown in the total antioxidant capacity between the unused and spent substrates, suggesting that the mushrooms take up and strongly retain these compounds. Untargeted metabolomic analysis revealed that many low-molecular-weight compounds, including polyphenols, were reduced in the SMS after cultivation, likely due to degradation or utilization by the fungal mycelium. Interestingly, SMSs from both mushroom species were found to be enriched in specific metabolites, particularly those associated with cerebrosides and ganoderenic acid D. These findings suggest potential for the high-value utilization of SMSs, including cosmetic or dermatological ingredient development, given their reported anti-inflammatory and moisturizing properties. However, further targeted analytical validation is required to confirm the identity and bioactivity of these enriched compounds. With continued advances in analytical and bioprocessing technologies, SMSs may also be more effectively characterized and valorized, enabling the selective recovery or enhancement of beneficial metabolites.

## Figures and Tables

**Figure 1 foods-15-00109-f001:**
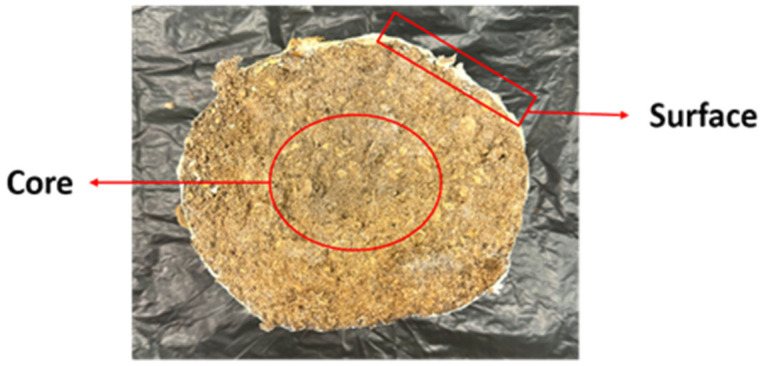
Cross section of the spent mushroom substrate; the surface and the core were collected for analysis.

**Figure 2 foods-15-00109-f002:**
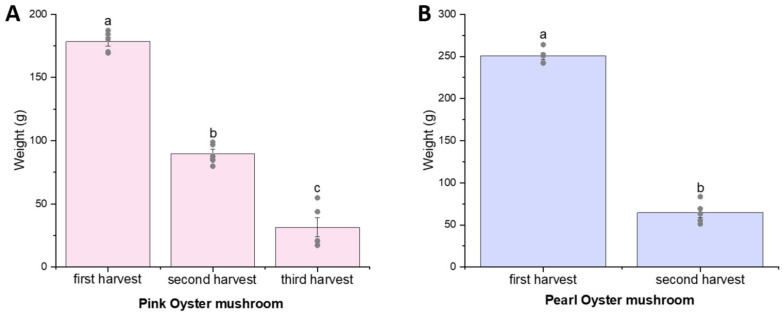
Yield of fruiting body for pink oyster (**A**) and pearl oyster (**B**) mushroom over multiple harvests. Values shown are mean ± S.E.M. (*n* = 5). Means in the same figure with different letters (a, b, c) were significantly different (*p* < 0.05, ANOVA, Bonferroni test).

**Figure 3 foods-15-00109-f003:**
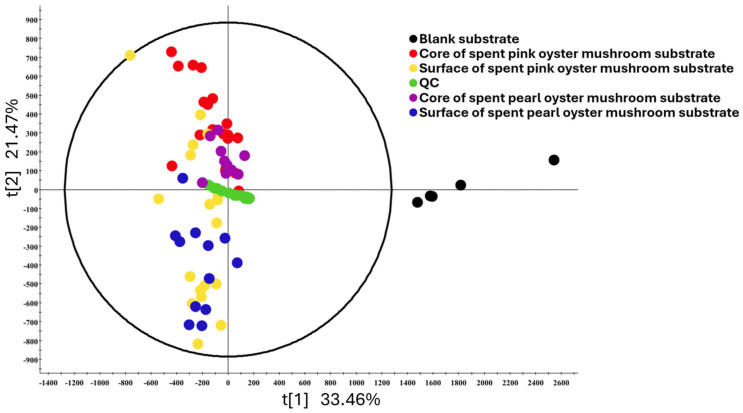
PCA of SMS and unused substrate (*n* = 5), grouped by different types of SMS.

**Figure 4 foods-15-00109-f004:**
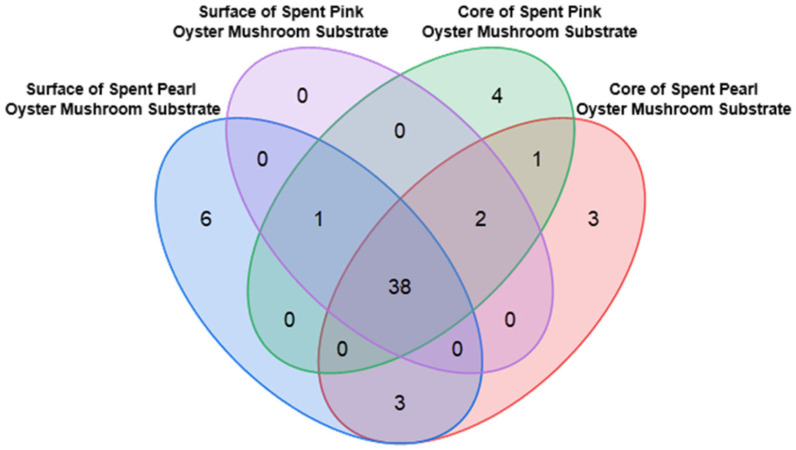
Overlap of compounds identified in different types of SMSs that show significant differences compared to the unused substrate.

**Figure 5 foods-15-00109-f005:**
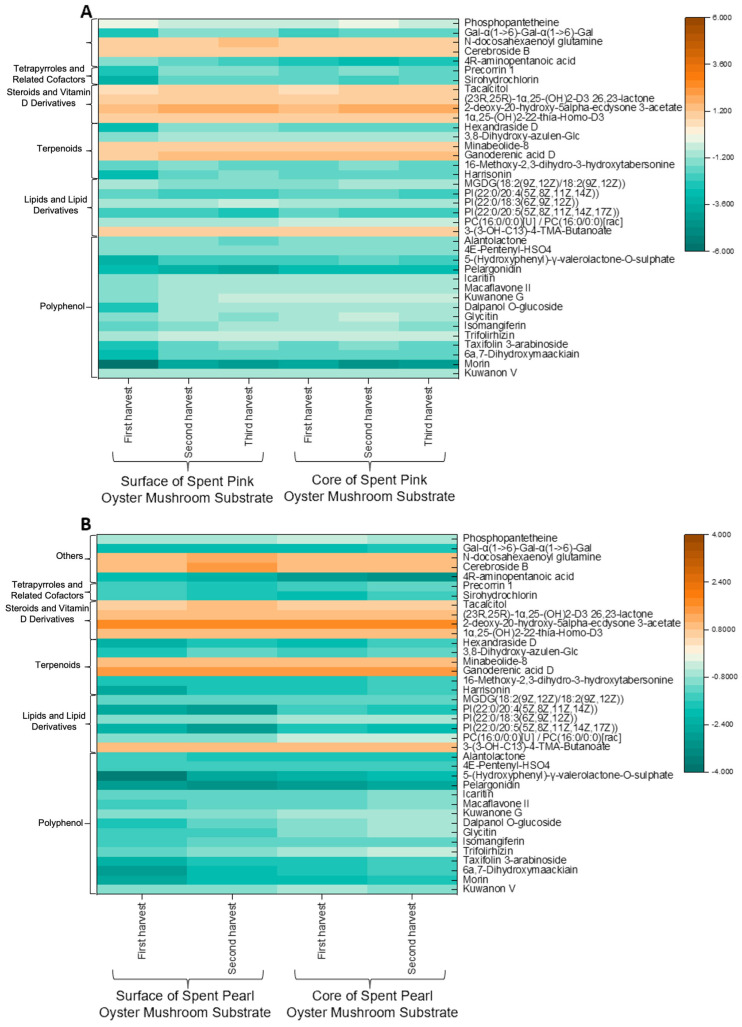
Heatmap of the Log_10_-transformed changes in putatively annotated metabolites across different harvests in pink oyster mushroom (**A**) and pearl oyster mushroom (**B**) compared to the unused substrate (*n* = 5). Gal-α(1->6)-Gal-α(1->6)-Gal: (1S,2S,3R,4R,5R,6S)-2,3,4,5,6-Pentahydroxycyclohexyl α-D-galactopyranosyl-(1->6)-α-D-galactopyranosyl-(1->6)-α-D-galactopyranoside; 3,8-Dihydroxy-azulen-Glc: 2-[(3S,3aR,5R,8R,8aS)-3,8-Dihydroxy-8-(hydroxymethyl)-3-methyl-2-oxodecahydroazulen-5-yl]propan-2-yl β-D-glucopyranoside; 4E-Pentenyl-HSO4: (4E)-5-(4-Methoxyphenyl)-2-methyl-3-oxo-4-penten-1-yl hydrogen sulfate; 3-(3-OH-C13)-4-TMA-Butanoate: 3-[(3-Hydroxytridecanoyl)oxy]-4-(trimethylammonio)butanoate; 1α,25-(OH)2-22-thia-Homo-D3: 1α,25-dihydroxy-24a-homo-26,27-dimethyl-22-thiavitamin D3/1α,25-dihydroxy-24a-homo-26,27-dimethyl-22-thiacholecalciferol; (23R,25R)-1α,25-(OH)2-D3 26,23-lactone: (23R,25R)-1α,25-dihydroxyvitamin D3 26,23-lactone/(23R,25R)-1α,25-dihydroxycholecalciferol 26,23-lactone.

**Table 1 foods-15-00109-t001:** EGT content in spent pink oyster mushroom substrate, spent pearl oyster mushroom substrate, and unused substrate.

	EGT Content (μg·g^−1^)				
	Pink Oyster Mushroom Substrate		Pearl Oyster Mushroom Substrate		Unused Substrate
	Surface of Substrate	Core of Substrate	Surface of Substrate	Core of Substrate	
First harvest	9.14 ± 1.13 ^a,A^	1.71 ± 0.44 ^a,B^	3.14 ± 0.45 ^a,B^	2.00 ± 0.57 ^a,B^	8.76 ± 0.66 ^A^
Second harvest	p.56 ± 2.65 ^a,A,B^	1.66 ± 0.26 ^a,B^	10.38 ± 1.56 ^b,A^	4.39 ± 0.72 ^b,A,B^	
Third harvest	4.53 ± 0.47 ^a,B^	2.03 ± 0.30 ^a,C^	-	-	

Values are mean ± S.E.M. (*n* = 5). Within each column, values with different lowercase superscript letters (a, b) are significantly different (*p* < 0.05, ANOVA, Bonferroni test), while within each row, values with different uppercase superscript letters (A, B, C) are significantly different (*p* < 0.05, ANOVA, Bonferroni test).

**Table 2 foods-15-00109-t002:** Content of EGT in pink oyster mushroom and pearl oyster mushroom fruiting body.

	EGT Content (μg·g^−1^)	
	Pink Oyster Mushroom	Pearl Oyster Mushroom
First harvest	782.62 ± 24.68 ^a,A^	760.03 ± 21.45 ^a,A^
Second harvest	738.30 ± 22.82 ^a,A^	519.13 ± 39.02 ^b,B^
Third harvest	1172.00 ± 82.73 ^b^	-

Values are mean ± S.E.M. (*n* = 5). Within each column, values with different lowercase superscript letters (a, b) are significantly different (*p* < 0.05, ANOVA, Bonferroni test), while within each row, values with different uppercase superscript letters (A, B) are significantly different (*p* < 0.05, ANOVA, Bonferroni test).

**Table 3 foods-15-00109-t003:** Contents of hercynine in spent pink oyster mushroom substrate, spent pearl oyster mushroom substrate, and unused substrate.

	Content of Hercynine (μg·g^−1^)				
	Pink Oyster Mushroom Substrate		Pearl Oyster Mushroom Substrate		Unused Substrate
	Surface of Substrate	Core of Substrate	Surface of Substrate	Core of Substrate	
First harvest	27.27 ± 5.20 ^a,A^	2.58 ± 0.14 ^a,B^	3.67 ± 0.97 ^a,B^	1.53 ± 0.48 ^a,B^	<0.05 nM
Second harvest	13.27 ± 3.59 ^a,b,A^	2.47 ± 0.98 ^a,B^	15.38 ± 0.79 ^b,A^	5.12 ± 0.27 ^b,B^	
Third harvest	4.76 ± 2.12 ^a,b^	1.02 ± 0.10 ^a^	-	-	

Values are mean ± S.E.M. (*n* = 5). Within each column, values with different lowercase superscript letters (a, b) are significantly different (*p* < 0.05, ANOVA, Bonferroni test), while within each row, values with different uppercase superscript letters (A, B) are significantly different (*p* < 0.05, ANOVA, Bonferroni test).

**Table 4 foods-15-00109-t004:** Contents of hercynine in pink oyster mushroom and pearl oyster mushroom.

	Content of Hercynine (μg·g^−1^)	
	Pink Oyster Mushroom	Pearl Oyster Mushroom
First harvest	835.29 ± 18.09 ^a,A^	760.03 ± 21.45 ^a,B^
Second harvest	738.30 ± 22.82 ^b,A^	543.96 ± 59.16 ^b,B^
Third harvest	150.11 ± 9.58 ^c^	-

Values are mean ± S.E.M. (*n* = 5). Within each column, values with different lowercase superscript letters (a, b, c) are significantly different (*p* < 0.05, ANOVA, Bonferroni test), while within each row, values with different uppercase superscript letters (A, B) are significantly different (*p* < 0.05, ANOVA, Bonferroni test).

**Table 5 foods-15-00109-t005:** Total phenolic contents (mg GAE·g^−1^ dry sample) in spent pink oyster mushroom substrate, spent pearl oyster mushroom substrate, and unused substrate.

	TPC (mg GAE·g^−1^)				
	Pink Oyster Mushroom Substrate		Pearl Oyster Mushroom Substrate		Unused Substrate
	Surface of Substrate	Core of Substrate	Surface of Substrate	Core of Substrate	
First harvest	2.30 ± 0.28 ^a,A^	1.20 ± 0.03 ^a,B^	1.25 ± 0.16 ^a,B^	1.12 ± 0.08 ^a,B^	1.58 ± 0.07 ^A,B^
Second harvest	1.88 ± 0.20 ^a,b,B,C^	1.11 ± 0.09 ^a,A,^	2.34 ± 0.12 ^b,C^	1.18 ± 0.04 ^a,A^	
Third harvest	1.43 ± 0.08 ^b,A^	0.84 ± 0.04 ^b,C^	-	-	

Values are mean ± S.E.M. (*n* = 5). Within each column, values with different lowercase superscript letters (a, b) are significantly different (*p* < 0.05, ANOVA, Bonferroni test), while within each row, values with different uppercase superscript letters (A, B, C) are significantly different (*p* < 0.05, ANOVA, Bonferroni test).

**Table 6 foods-15-00109-t006:** Total phenolic content (mg GAE·g^−1^ dry sample) in pink oyster mushroom and pearl oyster mushroom.

	TPC (mg GAE·g^−1^)	
	Pink Oyster Mushroom	Pearl Oyster Mushroom
First harvest	8.22 ± 0.18 ^a,A^	8.06 ± 0.07 ^a,A^
Second harvest	8.04 ± 0.08 ^a,A^	7.57 ± 0.11 ^b,B^
Third harvest	6.76 ± 0.32 ^b^	-

Values are mean ± S.E.M. (*n* = 5). Within each column, values with different lowercase superscript letters (a, b) are significantly different (*p* < 0.05, ANOVA, Bonferroni test), while within each row, values with different uppercase superscript letters (A, B) are significantly different (*p* < 0.05, ANOVA, Bonferroni test).

## Data Availability

The original contributions presented in this study are included in the article and Appendix A. Further inquiries can be directed to the corresponding author.

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
