# Peer review of "Foods2026, 15(1), 109;https://doi.org/10.3390/foods15010109"

_foods, 2025, doi:10.3390/foods15010109_

Round 1

Reviewer 1 Report

Comments and Suggestions for Authors

The manuscript “Characterization of Bioactive Compounds in Spent Mushroom Substrate: A Metabolomic Perspective on Its Untapped Potential” is well-structured, clearly written, and addresses an important sustainability and food science problem: the valorization of spent mushroom substrate (SMS) through metabolomics. The experimental design is generally solid, and the analytical approaches are appropriate and advanced.

Novelty

The authors claim that this work represents a breakthrough because it is the first time that ergotioneine (EGT) and its related metabolites have been systematically quantified in spent mushroom substrate (SMS), which is credible and represents a contribution. The work has acceptable scientific relevance because it addresses global issues such as agro-industrial waste, interest in bioactive compounds and the circular bioeconomy, food security, and the applications of functional ingredients.  It should be noted that the authors present a clear study objective and define the approach to waste utilization in the use of ergothioneine, comparing SMS and unused substrate, multiple collection cycles, and using fungal species. The experimental framework is well designed. It presents a solid and analytical approach combining LC-MS/MS specific for EGT, hercynin, and ETSO₃H. Non-specific LC-QTOF-MS metabolomics may have proposed a more robust methodology for antioxidant activity, showing CHNS elemental analysis results. These instrumentation-based results allow for conclusions to be drawn. For statistical analysis, the authors use a software package suitable for metabolomics work.  Among the findings highlighted in the study is the increase in terpenoids and vitamin D compounds.

Recomendaciones

Recommendations to be addressed, e.g., excessive reliance on putative identification

Many compounds in the non-specific section are considered identified (e.g., minabeolide, cerebrosides, vitamin D derivatives), but:

  • There is no MS/MS spectral match score.
  • Fragmentation patterns are not included.
  • Standards are not used for validation.

Currently, these compounds should be considered as “putatively annotated compounds” rather than confirmed identities.

Therefore, it is recommended to include in methods and discussion that the identities of the compounds were assigned presumptively based on exact mass and database matching. It is also important to include confidence levels according to the Metabolomics Standards Initiative and, where possible, validate at least 2-3 key compounds with standards.

There is a limited biological interpretation of the metabolome. The paper lists interesting compounds, but does not fully address key issues: Why is ganoderic acid found in the Pleurotus substrate? Possible source: contamination? Microbial activity? Lignins from wood? Why do vitamin D derivatives appear without ergocalciferol?

The paper presents some inconsistencies that need clarification. The authors state in the paper: SMS is considered an antioxidant source. However, their data show that the TPC is not significantly higher than the control. These seemingly contradictory results reflect differences in fungal species, substrate composition, extraction solvent polarity, and the degree of lignin degradation between studies.

Authors are kindly requested to review the grammar, as there are several sentences that need improvement, as well as the spelling of abbreviations and duplicate references (e.g., 45 and 46). Please also review Figures 3 (Add % variance under axes), 4 (Add legend for color scale), and 5 (Add sample size and category labels).

A suggested recommendation that is optional for authors is to include a brief paragraph on practical applications of SMS metabolites. This should show that the results of this study, although SMS may not be suitable for nutraceutical recovery of EGT, may represent an alternative raw material for vitamin D enrichment and triterpenoid extraction. Important for the development of cosmetic or dermatological ingredients. This can give the work a high value of applicability.

Comments on the Quality of English Language

The grammar should be reviewed by the authors to improve the manuscript overall, for example: 

It is recommended to change

“to increase in number” to “with an increase in number”, “best of our knowledge” to “To the best of our knowledge”, “vitamin D related derivatives” to “vitamin D-related derivatives”

Reviewer 2 Report

Comments and Suggestions for Authors

Dear authors,

Indeed, you have put a lot of work into this research paper and well drafted a interesting finding on mushroom substrates bioprospecting. Few points are highlighted in the manuscript for incorporation in the final draft for more clarity and understanding.

  • Line 82: how many bags of each mushroom was cultivated? please clarify
  • line 84-85: add reference 
  • line 87: mentioned name of model and manufacturer, if you are using brand equipment.
  • Line: 97: change word The structure of to Cross section of
  • line 101: start a sentence with words not by numerical 
  • line 103: mentioned name of model and manufacturer, if you are using brand equipment.
  • Line 130:  start sentence with full form not by abbreviation
  • Line 144-147: add a appropriate reference 
  • Line 150: start sentence with full form not by abbreviation
  • Line 160 : add a appropriate reference 
  • Line 121: change to increased 
  • line 125: change to Bodies 
  • Line 128: change to indicated
  • line 129: lack clarity the original substrate 
  • line: 222: what is unused substrate? How did you sampled it? collection and sample preparation is not mentioned in materials and Methods. 
  • Line 240: change is to was
  • Line 243: change word align to aligned 
  • line 269: change word "cultivation location" to "sites of collection/location"
  • line 306: change word shows to showed
  • line 314:" change word indicate to indicated 
  • line 336 change word are to were
  • line 365: expand word ECT
  • Line 365: re check whether fruiting bodies or substrate is used for analysis. lack clarity
  • line 395: conduct to conducted

General comment:

1. why physiochemical properties of substrates is not studied before cultivation of mushroom? so that results can be compared with post-cultivation data.

2. whether main constituent of substrate is only saw dust or included other supplement? need to check properly. This may impact on metabolomic studies of substrates. 

3. whether pink and oyster mushroom can be grown in same season? mentioned scientific name of these mushroom in sample collection section.

4. In conclusion, add more future prospect of SMS with advance technology

5. All the references need to be cross checked strictly following journal formats and guidelines

Reviewer 3 Report

Comments and Suggestions for Authors

Alvin Eng Kiat Loo et al., authors of the article entitled: “Characterization of Bioactive Compounds in Spent Mushroom Substrate: A Metabolomic Perspective on Its Untapped Potential”, focused on ergothioneine, a natural amino acid with anti-inflammatory and antioxidant properties. They measured the proportions of ergothioneine in spent oyster mushroom substrates, considering a potential form of recycling for this waste. Unfortunately, the remaining levels of ergothioneine are low due to degradation. However, in conducting this study, they were able to show an enrichment in compounds associated with vitamin D biosynthesis. Ergocalciferol (vitamin D₂) was not observed, but other derivatives were. Similarly, the presence of ganoderic acid derivatives, notably ganoderic acid D, a triterpenoid known for its anti-inflammatory and hepatoprotective effects, was also detected in these spent mushroom substrates. These derivatives are also of great interest, even though they were not the derivatives initially sought. Added to this is the presence of other interesting compounds, which encourages the authors to continue to pay particular attention to this waste in order to recover it and contribute to the circular economy. 
1)  Please note that line 421 reads “Especially ganoderenic acid D” instead of “Especially ganoderic acid D.”

 2) Similarly, I think the conclusion should be modified and expanded before publication, as it closely resembles the abstract. It should focus more on the results obtained. 

Reviewer 4 Report

Comments and Suggestions for Authors

The experimental article «Characterization of Bioactive Compounds in Spent Mushroom Substrate: A Metabolomic Perspective on Its Untapped Potential» presented for consideration is devoted to a topical area of research focused on potential bioactive compounds in spent mushroom substrate across different harvest cycles. The manuscript is clearly structured, and the material is presented logically and consistently. The introduction concisely and scientifically substantiates the relevance and feasibility of the study. The experimental design is appropriately planned for the research objectives. The authors have obtained a significant amount of material, which is well presented and graphically illustrated. The article meets all the journal's requirements.

I have only one comment. I would suggest reconsidering the emphasis on ergothioneine (EGT) in the stated objective of the study. It is inappropriate to single it out as the study aims to analyze a broader spectrum of bioactive compounds in SMS during different harvest cycles.

As an option:

This study aims to analyze the potential bioactive compounds in SMS across different harvest cycles of Pleurotus species.
